# Functional Recovery by Transplantation of Human iPSC-Derived A2B5 Positive Neural Progenitor Cell After Spinal Cord Injury in Mice

**DOI:** 10.3390/ijms26188940

**Published:** 2025-09-13

**Authors:** Yiyan Zheng, Xiaohui Chen, Ping Bu, Haipeng Xue, Dong H. Kim, Hongxia Zhou, Xugang Xia, Ying Liu, Qilin Cao

**Affiliations:** 1Center for Translational Science, Florida International University, 11350 SW Village Pkwy, Port St. Lucie, FL 34987, USA; yiyzheng@fiu.edu (Y.Z.); xiachen@fiu.edu (X.C.); haxue@fiu.edu (H.X.); hozhou@fiu.edu (H.Z.); xxia@fiu.edu (X.X.); 2Department of Environmental Health Sciences, Robert Stempel College of Public Health and Social Work, Florida International University, 11200 SW 8th Street, Miami, FL 33199, USA; 3Vivian L Smith Department of Neurosurgery, University of Texas Health Science Center at Houston, Houston, TX 77030, USA; ping.bu@uth.tmc.edu (P.B.); dong.h.kim@uth.tmc.edu (D.H.K.); 4Center for Stem Cell and Regenerative Medicine, The Brown Foundation Institute of Molecular Medicine for the Prevention of Human Diseases, University of Texas Health Science Center at Houston, 6431 Fannin Street, Houston, TX 77030, USA

**Keywords:** neural stem cells, spinal cord injury, hiPSC, neural repair, transplantation

## Abstract

Human induced pluripotent stem cells (hiPSCs) hold great potential for patient-specific therapies. Transplantation of hiPSC-derived neural progenitor cells (NPCs) is a promising reparative strategy for spinal cord injury (SCI), but clinical translation requires efficient differentiation into desired neural lineages and purification before transplantation. Here, differentiated hiPSCs—reprogrammed from human skin fibroblasts using Sendai virus-mediated expression of OCT4, SOX2, KLF4, and C-MYC—into neural rosettes expressing SOX1 and PAX6, followed by neuronal precursors (β-tubulin III^+^/NESTIN^+^) and glial precursors (GFAP+/NESTIN+). Both neuronal and glial precursors expressed the A2B5 surface antigen. A2B5+ NPCs, purified by fluorescence-activated cell sorting (FACS), proliferated in vitro with mitogens, and differentiated into mature neurons and astrocytes under lineage-specific conditions. Then, NOD-SCID mice received a T9 contusion injury followed by transplantation of A2B5+ NPCs, human fibroblasts, or control medium at 8 days post-injury. At two months, grafted NPCs showed robust survival, progressive neuronal maturation (β-tubulin III^+^→doublecortin^+^→NeuN^+^), and astrocytic differentiation (GFAP+), particularly in spared white matter. Transplantation significantly increased spared white matter volume and improved hindlimb locomotor recovery, with no teratoma formation observed. These results demonstrate that hiPSC-derived, FACS-purified A2B5+ NPCs can survive, differentiate into neurons and astrocytes, and enhance functional recovery after SCI. This approach offers a safe and effective candidate cell source for treating SCI and potentially other neurological disorders.

## 1. Introduction

Spinal cord injury (SCI) is a devastating condition that often results in permanent motor and sensory deficits, with current treatments—surgical stabilization, pharmacological management, and rehabilitation—providing only limited functional recovery [1,2,3]. Stem cell-based therapies have emerged as a promising approach for promoting neural regeneration and restoring function [4,5,6,7,8]. Preclinical studies demonstrate that transplantation of human neural stem or progenitor cells (NSCs/NPCs) into injured spinal cords can enhance tissue repair, axonal regeneration, and behavioral outcomes [9,10,11,12,13,14,15].

Human NSCs/NPCs can be derived from fetal brain [10,16] and spinal cord [17,18], human embryonic stem cells (hESCs) [11,19,20,21], or human induced pluripotent stem cells (hiPSCs) [13,22,23,24]. While each of these cell sources has distinct advantages and limitations, hiPSCs have gained particular attention due to their ethical acceptability, immunological compatibility, and scalable differentiation into diverse neural lineages [25,26]. hiPSCs can be generated from patient-derived somatic cells using transient expression of OCT4, SOX2, KLF4, and C-MYC [27], with Sendai virus vectors offering an efficient, non-integrating, and clinically translatable reprogramming method [28,29,30,31]. Coupled with cGMP-grade reagents and standardized neural differentiation protocols, hiPSCs represent a viable, regulatory-compliant cell source for SCI therapy.

A major challenge for clinical translation is obtaining a lineage-committed NPC population with minimal contamination from undifferentiated or off-target cells, which can compromise efficacy and pose tumorigenic risks [32,33,34]. Surface marker-based sorting provides a clinically applicable alternative to reporter-based purification. The A2B5 monoclonal antibody, which recognizes a glycoganglioside specific to early glial and neural progenitors [35,36], has been widely used to isolate glial-restricted precursors from fetal and postnatal rodent and human central nervous tissues [37,38]. We have previously developed a fluorescence-activated cell sorting (FACS) method using A2B5 to purify NPCs from neural differentiated hiPSCs [30]. Our study has shown that the purified A2B5+ NPCs can survive and differentiate into both neurons and astrocytes after transplantation into the injured spinal cord [30], showing robust survival and neuronal and astrocytic differentiation after transplantation into injured spinal cords. However, their impact on functional recovery remains unknown, representing a critical gap for translational advancement.

In the present study, we generated human iPSC-derived neural progenitor cells (NPCs) by reprogramming dermal fibroblasts with Sendai viruses and directing differentiation under defined conditions. A2B5+ NPCs were purified by FACS, yielding a clinically relevant, lineage-specific population with robust proliferation and neuronal/glial differentiation in vitro. In a mouse contusion SCI model, transplanted A2B5+ NPCs survived, integrated into host tissue, and differentiated into appropriate neural lineages, leading to significant motor function recovery. This result represents the first behavioral evidence of efficacy for A2B5-purified hiPSC-derived NPCs in SCI.

## 2. Results

### 2.1. Human iPSC Reprogramming and Neural Differentiation In Vitro

Human fibroblasts were successfully reprogrammed into hiPSCs using non-integrating Sendai virus vectors encoding the reprogramming factors OCT4, SOX2, KLF4, and c-MYC, as described previously [30]. The hiPSCs were then induced into neuroectodermal lineages using our established NPC differentiation protocol [30]. Fifteen days post-differentiation, the cells formed typical neural rosettes that robustly expressed early NPC markers SOX1 and PAX6 (Figure 1A–C), indicating efficient neural induction. By day 26, numerous cells expressed NESTIN and were immunoreactive to A2B5 antibody (Figure 1D–F). In addition, they uniformly expressed β-tubulin III, a classic marker for neuronal progenitor cells and immature neurons (Figure 1G–I), reflecting further commitment along the neural lineage. By day 40, while still maintaining immunoreactivity to A2B5, cells had started expressing glial fibrillary acidic protein (GFAP), a marker of astrocytes (Figure 1J–L), suggesting the onset of gliogenesis. Taken together, our results showed that A2B5 is expressed in NESTIN+ NPCs+, β-tubulin III+ neuronal progenitor cells, and GFAP+ glial precursor cells or immature astrocytes. This is consistent with A2B5 being a marker of early neural lineage commitment. These findings demonstrate the stepwise and temporally regulated neural differentiation potential of the hiPSC-NPCs and confirm the presence of A2B5-positive progenitor populations across multiple neural stages.

### 2.2. Purification and Differentiation of hiPSC-Derived Neural Progenitor Cells In Vitro

To enrich for neural lineage cells, A2B5-positive NPCs were isolated by FACS (Figure 2A,B). Only the top 15% with the highest A2B5 expression were collected for subsequent experiments to ensure purity. Following purification, almost all of the sorted population were A2B5-positive at one day post-sorting (Figure 2C). These purified A2B5+ NPCs remained proliferative in vitro when cultured in the presence of bFGF, as shown by their continued expansion and expression of NESTIN (Figure 2D–F). Within ten days following withdrawal of bFGF and initiation of differentiation conditions, the A2B5+ NPCs continued to mature along neuronal and glial lineages, differentiating into β-tubulin III+ neuronal cells or GFAP+ astrocytes. However, at this point, the differentiated GFAP+ cells showed immature astrocyte morphology, exhibiting a simple, less-branched morphology with small cell bodies and few, short processes (Figure 2G–I). Continued differentiation under lineage-specific conditions further promoted maturation. After three weeks in neuronal differentiation medium, cells expressed mature neuronal markers MAP2 and neurofilament medium chain (NFM), indicating successful progression toward a more mature neuronal phenotype (Figure 2J,K). In parallel, cells maintained in astrocyte differentiation medium for four weeks developed into GFAP+ mature astrocytes, with a more complex and elaborate morphology characterized by larger somas and extensively branched, fine processes that radiated outward in a star-shaped pattern (Figure 2L). These results confirm that A2B5+ NPCs represent a proliferative and multipotent population capable of differentiating into both neuronal and glial lineages in vitro.

### 2.3. Survival, Integration, and Migration of Grafted iPSC-Derived Neural Progenitor Cells

To evaluate the survival and integration of grafted NPCs in long-term survival animals, we pre-labeled the A2B5+ NPCs with lentiviruses expressing EGFP and looked at their change in morphology. In short-survival animals (one week after grafting), unlabeled A2B5+ cells and human nuclei (hN) antibody were used to detect the survival and distribution. A2B5+ iPSC-NPCs were transplanted into the injured epicenter at eight days after T9 moderate contusion. At one week post-transplantation, hN-labeled NPCs were detected within the injury epicenter, indicating initial survival of the grafted cells (Figure 3A–C). These cells were predominantly localized within the injured regions surrounded by GFAP+ reactive astrocytes. At two months post-transplantation, robust survival of the grafted cells was observed in the injured spinal cord, as evidenced by co-labeling of human nuclei and GFP (Figure 3D,E). While a subset of NPCs remained within the injury core, a significant proportion migrated into the spared tissue surrounding the lesion. Notably, many surviving grafted cells were positive for human nuclei but lacked detectable GFP expression. This discrepancy is likely attributable to incomplete lentiviral GFP labeling prior to transplantation and possible downregulation of GFP during differentiation and maturation in vivo (Figure 3).

### 2.4. Differentiation of Grafted NPCs Following SCI

The grafted hiPSC-derived NPCs exhibited progressive neuronal differentiation over time following transplantation. At one week post-transplantation, a substantial number of grafted human cells, labeled by human nuclei staining, expressed β-tubulin III, indicative of a neuronal progenitor phenotype (Figure 4A–C). By two weeks post-graft, these cells began to express doublecortin (DCX), a marker of immature neurons (Figure 4D–F), suggesting ongoing neuronal lineage commitment. At two months post-graft, the majority of grafted A2B5+ NPCs had differentiated into mature neurons expressing NeuN, primarily localized in the spared gray matter adjacent to the injury site. Notably, GFP^+^ processes from graft-derived neurons extended into the spared ventral horn, where some were positioned in close proximity to host motor neurons and interneurons (Figure 4G–I). These findings suggest the potential for grafted neurons to form connections with host neurons. However, further studies are required to validate synaptic integration of grafted neurons into host circuits, for example, using immuno-electron microscopy or transsynaptic viral tracing. In addition to neuronal differentiation, a subset of grafted A2B5+ NPCs adopted a glial fate. Specifically, some cells differentiated into GFAP+ astrocytes located in the spared spinal cord surrounding the injury area (Figure 4J–L), indicating the potential of grafted NPCs to contribute to both neuronal and astroglial populations in the injured spinal cord.

### 2.5. Preservation of White Matter Following Transplantation of A2B5+ iPSC-NPCs in SCI

To assess the neuroprotective effects of A2B5+ iPSC-NPCs transplantation on white matter (WM) integrity after injury, we quantified spared WM areas and volumes in the injured spinal cord using eriochrome cyanine (EC) staining at two months post-transplantation. In the normal T9 spinal cord, EC-positive WM surrounds the H-shaped, largely EC-negative gray matter (Figure 5A). Following SCI, EC-positive WM was markedly reduced at the lesion epicenter, indicating the greatest WM loss at this site (Figure 5B). Spared WM gradually increased with distance from the epicenter both rostrally and caudally (Figure 5B). Quantitative analysis revealed that significantly greater spared WM areas from the epicenter to 2.4 mm rostrally and caudally in animals receiving NPC grafts compared to either human fibroblasts or culture medium controls (Figure 5C). Total spared WM volume was also significantly higher in the NPC-grafted group than in both control groups (Figure 5D). These results suggest that hiPSC-derived NPCs contribute not only to cellular replacement but also to structural preservation of the spinal cord following traumatic injury.

### 2.6. Functional Recovery After Transplantation of Human iPSC-Derived NPCs After SCI

To evaluate if transplantation of A2B5+ iPSC-NPCs could improve the functional recovery following SCI, locomotor function was assessed using the Basso Mouse Scale (BMS), BMS sub-scores, and beam-walking tests in mice that received A2B5+ hiPSC-NPC grafts. Grafting of human fibroblasts or injection of culture medium was used as controls. BMS scores (Figure 6A) were significantly higher in the NPC-transplanted group compared to the human fibroblasts grafting group starting at 3 weeks post-injury (2 weeks after transplantation), indicating early improvement in gross locomotor function. From week 4 post-injury onward, mice in the NPC group demonstrated significantly greater BMS scores compared to both human fibroblasts and medium control groups, suggesting sustained functional benefits of A2B5+ NPC transplantation. Similarly, BMS sub-scores (Figure 6B), which offer a more detailed analysis of specific aspects of movement during different phases of recovery, such as ankle movement, paw placement, trunk stability, and coordination, were significantly higher in the NPC group compared to both control groups beginning at 5 weeks post-injury. In the beam-walking test (Figure 6C), which measures balance and coordination on a narrow beam, NPC-transplanted mice showed significantly improved scores compared to the human fibroblasts group at 5 weeks post-injury, with further improvements observed at 6 weeks post-injury and beyond when compared to both control groups. These results demonstrate that transplantation of A2B5+ hiPSC-derived NPCs promotes significant and sustained improvements in locomotor function in mouse SCI models.

## 3. Discussion

In the present study, we successfully generated, differentiated, and purified hiPSC-derived NPCs marked by A2B5 immunoreactivity. To assess their in vivo therapeutic potential in the context of SCI, we transplanted these NPCs into the injured spinal cords of NOD-SCID mice subjected to a T9 contusion injury. Following transplantation into T9 contusion-injured NOD-SCID mice, NPCs survived, integrated, migrated from the lesion core into spared tissue, and differentiated into neurons and astrocytes. NPC grafts promoted substantial white matter preservation and improved locomotor recovery, shown by BMS scores, sub-scores, and beam-walking. These functional improvements emerged by two weeks and persisted throughout the study. These findings highlight the therapeutic potential of hiPSC-derived NPCs for SCI repair.

Our results demonstrated that transplantation of NPCs derived from A2B5+ hiPSCs promoted the recovery of locomotor function after SCI. These findings are consistent with previous studies examining the therapeutic potential of hiPSC-derived NPCs using a variety of SCI models, including mouse thoracic contusion [24,39,40,41,42,43], rat thoracic contusion or compression [44,45,46,47,48], rat cervical hemisection [22], and non-human primate cervical contusion [14]. Collectively, these studies provide strong preclinical evidence supporting the translation potential of hiPSC-NPCs for SCI treatment. However, tumor-like overgrowth of grafted hiPSC-NPCs was reported in several studies [32,40,49,50], raising significant safety concerns for their clinical application. The risk of tumorigenesis remains one of the major barriers to clinical translation of hiPSC-NPCs. Various strategies have been explored to mitigate this risk. For example, pretreatment of NPCs with gamma secretase inhibitors prior to transplantation has been shown to reduce proliferation and promote differentiation [40,42]. In cardiac research, a glucose-free medium has been used to metabolically select cardiomyocytes from hiPSCs [51,52]. However, it remains unclear whether this approach would be effective for purifying NPCs from hiPSCs, as proliferative NPCs may also be glucose-dependent. Overall, these approaches may not fully eliminate the tumorigenic cells. Inducible suicide genes have also been explored as a safety mechanism to ablate the tumors post-transplantation [49,53]. However, these approaches cannot prevent tumorigenesis and require genetic modification of the hiPSCs, which introduces additional safety concerns. In this study, we tested the safer alternative of transplantation of FACS-purified A2B5+ NPCs. These cells promoted locomotor recovery without any evidence of tumor formation. Importantly, A2B5+ sorting effectively eliminated undifferentiated hiPSCs [30], avoiding the risk of teratoma formation. Beyond teratoma, overgrowth of immature NSCs can also contribute to tumorigenesis. Previous studies [32,49,53] have reported initial improvement in locomotor function during the first 6 weeks after transplantation of hiPSCs-NSCs/NPCs, followed by abrupt deterioration due to tumor growth. Some subpopulations of immature NSCs may remain undifferentiated and continue proliferating after transplantation, driving tumor formation in murine SCI models.

In contrast, our study showed significant locomotor improvement within the first 6 weeks post-transplantation of A2B5+ NPCs, with the recovery either sustained (as measured by BMS) or slightly improved (beam-walking score) during week 7–9. Histological analyses confirmed the absence of tumor formation in all animals that received A2B5+ NPC grafts up to 2 months post-transplantation. Our in vitro studies revealed that A2B5 was not expressed in the highly proliferative NSCs located at the center of neural rosettes but is instead present in more mature NPCs surrounding the rosette center. The purified A2B5+ NPCs could remove these highly proliferative NSCs and prevent the tumor formation due to overgrowth of grafted immature NSCs. Thus, FACS purification of A2B5+ NPCs likely eliminates undifferentiated hiPSCs and highly proliferative NSCs, reducing the risk of tumor formation from either source. Taken together, our results suggest that A2B5+ NPCs represents a safer cell source for SCI transplantation. Nevertheless, further studies are needed to identify markers for tumorigenic NSCs in the injured spinal cord after transplantation and clarify their relationship with the highly proliferative A2B5 negative NSCs. In addition, long-term studies are essential to further validate the safety of A2B5+ NPC transplantation.

Although the precise mechanisms by which A2B5+ NPC transplantation promotes functional recovery remain yet to be fully elucidated, our results suggest that several complementary mechanisms may be involved. We observed a progressive maturation of grafted A2B5+ NPCs, transitioning from neuronal progenitor cells to immature neurons and then to mature neurons over the course of 1 to 8 weeks post-transplantation. Notably, graft-derived neurons extended processes into the spared ventral horn where some were positioned in close proximity to host motor neurons and interneurons. These findings are consistent with previous studies [54,55] indicating that the grafted neurons could be potentially integrated into host neuronal circuits in the ventral horn and contributed to the locomotor recovery. In addition to neurons, our results demonstrated that grafted A2B5+ NPCs also differentiated into astrocytes. These astrocytes may provide a permissive environment for the regeneration of injured axons into the graft [56,57]. Furthermore, astrocytes derived from the graft may support the survival of co-grafted neurons and facilitate their integration into host circuits. Thus, neurons and astrocytes generated from the NPC graft may act synergistically to enhance circuit integration and contribute to functional recovery. Moreover, we observed that A2B5+ NPC transplantation significantly increased spared white matter around the injury, suggesting a neuroprotective role in reducing myelin and axonal loss after SCI. Grafted NPCs may exert neuroprotective effects by producing and secreting multiple growth factors, neurotrophins, and cytokines within the injured spinal cord microenvironment [58]. For example, grafted NPCs upregulate the expression of neurotrophins such as NGF, BDNF, and GDNF in the injured cervical spinal cord, which can reduce axonal loss and promote outgrowth [59]. Increased production of additional trophic factors, including CNTF and IGF-1, has also been reported following NPC transplantation in a rat clip compression SCI model [60]. Grafted NPCs may further mitigate tissue damage and promote recovery by modulating immune responses through the releasing of various chemokines and cytokines to regulate macrophage and microglial phenotypes and inhibit antigen presentation by dendritic cells [61,62]. In summary, A2B5+ NPC transplantation may contribute to locomotor recovery through a combination of neuronal and astrocytic differentiation and neuroprotective effects. Future gain- or loss-of-function studies will be essential to delineate the relative contribution of each mechanism. These investigations will provide critical insights into how A2B5+ NPC transplantation promotes recovery after SCI and will inform the development of optimized NPC-based therapies.

One potential limitation of our study is that the purified A2B5+ NPCs were not specialized to a spinal cord identity. Although most previous studies have used non-spinal cord-committed NSCs or NPCs and still demonstrated functional benefits after transplantation in various SCI models [15,22,41], caudalized human NPC grafts with a spinal cord fate have been shown to promote corticospinal tract regeneration after SCI more effectively than rostralized ones with forebrain or hindbrain identity [57]. Protocols to generate human spinal cord NPCs from pluripotent stem cells have been established [63,64]. Future studies are needed to directly compare the therapeutic efficacy of A2B5+ NPCs with or without spinal cord specialization in the clinically relevant contusion SCI models. Such studies will help to determine the optimized NPC source for SCI treatment.

In summary, NPCs purified from hiPSCs using A2B5-based FACS survived, differentiated into neurons and astrocytes, reduced injury size, and improved functional recovery after SCI without macroscopic overgrowth and teratomas formation for up to 2 months. With clinical trials of hiPSC-derived NPCs for SCI underway [65,66] but lacking safety and efficacy data, our findings suggest that incorporating A2B5-based FACS into cGMP-compliant protocols could yield safer, clinical-grade NPCs. Using A2B5 as a single marker streamlines purification, lowers costs, and facilitates regulatory approval, providing a practical advantage over multi-marker sorting strategies for clinical translation.

## 4. Materials and Methods

### 4.1. Human iPSC Reprogramming

Dermal fibroblasts from healthy individuals were reprogrammed into human induced pluripotent stem cells (hiPSCs) using the CytoTune-iPS 2.0 Sendai Reprogramming Kit (Thermo Fisher Scientific, Waltham, MA, USA), following the manufacturer’s protocol. Briefly, fibroblasts were plated at a density of approximately 5 × 10^4^ cells per well in a 6-well plate and allowed to reach ~80% confluency. Cells were then transduced with Sendai viral vectors encoding the four reprogramming factors OCT4, SOX2, KLF4, and c-MYC at the recommended multiplicity of infection (MOI). After 24 h, the medium was replaced, and cells were maintained in fibroblast medium for an additional 4–5 days. On day 5 post-transduction, cells were transferred onto matrigel-coated plates (Corning, Corning, NY, USA, Cat. No. 08-774-552) and cultured in mTeSR Plus medium (Stem Cell Technologies, Vancouver, BC, Canada, Cat. No. 100-0276). Emerging iPSC colonies with ESC-like morphology were manually picked between days 18–25 and expanded for downstream characterization. Cells were examined by immunofluorescence staining for pluripotent markers. Karyotyping analysis was performed every 10 passages to confirm that the iPSCs maintained a normal karyotype.

### 4.2. Cell Culture

Human iPSC lines were maintained under defined, feeder-free conditions in mTeSR Plus medium (Stem Cell Technologies, Cat. No. 100-0276). Cells were routinely passaged every 4 to 5 days using Accutase (Innovative Cell Technologies, San Diego, CA, USA, Cat. No. AT104) and replated at a 1:4 to 1:8 ratio onto matrigel-coated plates (Corning, Cat. No. 08-774-552) in the presence of 10 µM ROCK inhibitor Y-27632 to enhance cell survival, following the manufacturer’s guidelines and previously established protocols [13].

### 4.3. Differentiation of iPSCs into Neural Progenitor Cells (NPCs)

Human iPSCs were digested with 0.5 mM EDTA at 37 °C for 10 min and gently broken into small clumps. These clumps were transferred to non-adherent Petri dishes (Corning) and cultured in iPSC medium lacking basic fibroblast growth factor (bFGF) for 8 days to promote embryoid body (EB) formation, following established protocols [13]. The resulting EBs were then plated onto adherent culture plates and transitioned to neural induction medium consisting of DMEM/F12 with GlutaMAX, 1× non-essential amino acids (NEAA), 1× N2 supplement, and 20 ng/mL bFGF. After 2–3 days in neural induction conditions, neural rosettes began to emerge. These rosettes were manually selected, dissociated into single cells, and expanded in Neurobasal medium supplemented with 1× NEAA, 2 mM L-Glutamine, 1×B27 supplement, and 20 ng/mL bFGF. Unless otherwise specified, all reagents were purchased from Thermo Fisher Scientific.

### 4.4. Purification of A2B5+ NPCs by FACS

A2B5+ NPCs were purified by FACS as described previously [30]. Around 30 days after neural differentiation of iPSCs, cells were detached from cultured dishes by Accutase and collected by centrifugation. The cell pellets were then resuspended in 1× phosphate-buffered saline (PBS) containing 0.5% fetal bovine serum (FBS) at a concentration of 10^7^ cells/mL. Cells were then stained with primary antibody (anti-A2B5 mouse IgM), followed by a FITC-conjugated secondary antibody, and purified using a FACSAria II cell sorter system (BD) at 4 °C, at a rate of 2500 cells/s. The purified A2B5+ NPCs were seeded onto matrigel-coated plates in growth medium, which consisted of DMEM-F12, N2 and B27 supplements, and bFGF (20 ng/mL). Growth medium was changed every other day and bFGF2 was added daily. The cells were passaged at 60–70% confluence. In all cases, an aliquot of cells was analyzed by FACS and immunostaining to determine the efficiency of FACS. Only those cell preparations in which >95% of the sorted cells stained positive for A2B5 were used in the experiments.

### 4.5. In Vitro Differentiation of NPCs

To differentiate NPCs into neurons in vitro, the cells were plated on matrigel-coated 24-well plates in Neurobasal medium supplemented with 1× NEAA, 2 mM L-Glutamine, 1× B27 supplement, 1 ng/mL bFGF, 10 ng/mL NT3, 10 ng/mL BDNF and 10 ng/mL GDNF. The culture medium was changed every three days. After 1 and 3 weeks after differentiation, the cells were fixed and immunohistochemically analyzed for the neuronal markers, β-tubulin III, MAP2, b and NFM. To differentiate NPCs into astrocyte in vitro, the cells were seeded on matrigel-coated 24-well plates in culture medium which consisted of DMEM-F12, 1× N2 and 1× B27 supplements, and 2 ng/mL bFGF and 20 ng/mL bone morphogenetic protein 4 (BMP4). The medium was changed every three days. After 1, 2, or 4 weeks, the cells were fixed and immunohistochemically analyzed for the astrocyte marker, GFAP.

### 4.6. Cell Preparation for Transplantation

To facilitate tracking of grafted cells, some groups received A2B5+ NPCs labeled with a lentivirus encoding enhanced green fluorescent protein (EGFP) driven by the constitutive CMV promoter (Appendix A), while others received unlabeled cells prior to transplantation. For EGFP lentiviral infection, NPCs were treated with 1 μg/mL polybrene for 1 h, followed by incubation in growth medium containing lentiviruses expressing EGFP for 4 h. Routinely, about 60% cells were labeled. Two hours before transplantation, the labeled or unlabeled NPCs were detached from the dishes by Accutase, collected by centrifugation at 300× *g* for 5 min, and resuspended in 1 mL culture medium. After cell count and viability assessment with trypan blue in a hemacytometer, the cell suspension was centrifuged a second time and resuspended in a smaller volume to give a density of 5 × 10^4^ viable cells/μL.

### 4.7. Animals

All animal care and surgical procedures were conducted in strict accordance with protocols approved by Animal welfare committee at UTHealth (protocols AWC-14-0167, AWC-18-0156). Adult NOD-SCID mice were purchased from the Jackson Laboratory (NOD.Cg-Prkdc scid/J, strain no. 001303) and housed under a 12 h light/dark cycle with ad libitum access to food and water in pathogen-free animal facilities. All studies were conducted using female NOD-SCID mice aged 3–6 months. In studies with behavioral tests, mice were acclimated to the tests for three days and then had baseline BMS and beam-walk tests before injury. Only animals with the normal BMS score (9) and beam-walk score (25) were used. These mice were randomly allocated to three groups (NPC grafts, human fibroblast control (hFB ctr), and medium control) at 1:1:1 in a blinded manner using the Randbetween function in Excel. Experimenters will be blinded to treatments during experiments and data analysis. Animal numbers will be coded during behavioral and histological analysis. Animals from different groups will be housed together. The sample size was determined by a priori power analysis using G*power software (3.19.4) (N = 16 per group, 3 groups total, based on alpha value of 0.01 (chosen conservatively), 0.5 effective size, 0.7 correlation among repeated measures resulting in a power of 0.83). The number of mice used for behavioral tests and histology assessments was outlined in Table 1. In addition to mice used for behavioral tests, 8 mice were used for evaluating the survival and differentiation of grafted NPCs at 1 week or 2 weeks after transplantation with 4 mice in each time point.

### 4.8. Thoracic Contusion SCI and Cell Transplantation

Surgical techniques were adapted from previously published methods [30,67] with minor modifications. Adult female NOD-SCID mice were anesthetized using ketamine (80 mg/kg), dexmedetomidine (0.25 mg/kg), and butorphanol (0.5 mg/kg), or with isoflurane inhalation. A dorsal laminectomy was performed at the thoracic level 9 (T9) to expose the spinal cord. The spine was stabilized during the procedure using the steel stabilizers placed beneath the transverse processes of the adjacent vertebrae. A moderate central contusion injury (60 kdyne) was then induced using an Infinite Horizons (IH) impactor (Infinite Horizons LLC, Lexington, KY, USA). Following contusion, surgical wounds were closed in layers, and bacitracin ointment (Qualitest Pharmaceuticals, Huntsville, AL, USA) was applied topically. Mice received a subcutaneous injection of gentamicin (0.1 mL of 2 mg/mL, ButlerSchein, Dublin, OH, USA) and were allowed to recover on a water-circulating heating pad. After injury animals were monitored, they were fed with pre-softened food more palatable and received prophylactic against pain. Postoperative care included subcutaneous administration of buprenorphine (0.05 mg/kg, twice daily for three days; Reckitt Benckiser, Hull, UK) and manually emptying the bladder twice a day until the automatic voiding returned spontaneously.

At 8 days after SCI, the NOD-SCID mice were randomly assigned into three groups to receive graft of DMEM, human fibroblasts, or human iPSC-derived A2B5+NPCs, respectively. We did not include an unsorted NPC group, as prior studies have shown that unsorted NPC grafts frequently lead to tumor-like overgrowth beginning around 6 weeks post-transplantation [32,40,49,50]. Similarly, we excluded the A2B5-negative group for two reasons: (1) it contains mixed populations, including undifferentiated iPSCs that inevitably cause tumor formation after grafting; and (2) the heterogeneity of these cells makes it difficult to directly compare therapeutic efficacy with the lineage-restricted A2B5+ NPCs. Therefore, we focused on characterizing A2B5-positive sorted cells for grafting and subsequent experiments. The injured mice were re-anesthetized as above, the original laminectomy site was re-exposed, and 2 μL of cell suspension (total of 10^5^ cells) were injected into the injury epicenter at a depth of 0.7 mm using a glass micropipette (outer diameter 50–70 μm, tip beveled to 30–50°), at a controlled rate of 0.2 μL/min, for a total dwell time of 5 min, as previously described [68]. Mice were monitored postoperatively and allowed to survive for 8 weeks following transplantation for further analysis.

### 4.9. Behavioral Assessments

#### 4.9.1. Basso Mouse Scale (BMS)

Mice were acclimated to behavioral testing for three days prior to SCI. Baseline and post-injury motor function were assessed weekly for 9 weeks using the Basso Mouse Scale (BMS) score and BMS sub-score. To ensure unbiased assessment, animals were randomly coded, and all behavioral testing and scoring were performed by two independent investigators blinded to treatment groups. The BMS locomotor scoring system was used to evaluate hindlimb motor recovery following SCI, as previously described [69]. Mice were placed in an open field and observed for 4 min. Scores range from 0 (no observable hindlimb movement) to 9 (normal locomotion), incorporating criteria such as joint movement, plantar stepping, coordination, trunk stability, and tail position. Both hindlimbs were scored separately and averaged for analysis.

#### 4.9.2. Elevated Gradient Beam-Walking

The beam-walking test was conducted as described previously [70,71]. Briefly, the mice were evaluated on a graded series of metal beams (25 cm in length) of various widths: 0.4, 0.8, 1.2, 1.6, 2 cm. The beams were suspended (15 cm high) across a plastic box filled with 5–6 cm of soft bedding. The beam-walking score, range from 0 to 25, was calculated by the major score in the narrowest beam the mice crossed minus the average hindlimb missteps every run while crossing the beam four times. The major score in 2, 1.6, 1.2, 0.8, and 0.4 cm width beam was 5, 10, 15, 20, and 25, respectively. A beam-walking score of 0 indicated the animal’s inability to stand on the beam or dragging its hindlimbs on a 2 cm wide beam without body support, and a score of 25 revealed that the animal was able to walk across a 0.4 cm beam four times without errors. The narrowest beam each mouse could traverse was recorded, and the number of missteps over four trials were counted and averaged. Hindpaw and whole-body falls were both counted as missteps. If an animal could not maintain placement of its hindpaws on the beam, or if the animals were dragging its hindlimbs across the beam, this was considered failing the task and 0 was scored for the animal.

Weekly BMS scores and sub-score as well as the beam-walking score were averaged across animals in each treatment group and plotted over time. Statistical analysis was performed using two ways ANOVA followed by Tukey’s HSD post hoc tests to determine group differences at each time point across the 9-week testing period.

### 4.10. Immunofluorescence and Immunohistochemistry

Mice were anesthetized with a mixed solution of ketamine (80 mg/kg), dexmedetomidine (0.25 mg/kg), and butorphanol (0.5 mg/kg), and perfused transcardially with 0.01 M phosphate-buffered saline (PBS, pH 7.4), followed by 4% paraformaldehyde (PFA) in PBS. The injured spinal cord segments were removed, post-fixed in 4% PFA overnight, cryoprotected in 20% sucrose overnight, 30% sucrose overnight at 4 °C, and embedded in OCT compound (Fisher Scientific). The cords were cryosectioned in 20 µm slices either transversely or longitudinally and mounted serially on Superfrost Plus Gold Slides.

For immunofluorescence staining, slides were blocked with 10% donkey serum in Tris buffered saline (TBS) containing 0.2% Triton X-100 (TBST) for 1 h at RT. The sections were then incubated in primary antibodies diluted in blocking buffer at 4 °C overnight. Appropriate secondary antibodies were used for single and double labeling. All secondary antibodies were tested for cross-reactivity and nonspecific immunoreactivity. The following primary antibodies were used: A2B5 (1:20, ATCC, cat. no. CRL1520), β tubulin III (1:1000, Sigma, cat. no. MAB1637), DCX (doublecortin, 1:200, Abcam, cat. no. 18723), GFAP (1:2000, DAKO, cat. no. Z0334), GFP (1:200, Abcam, cat. no. ab13970), human nuclei (hN, 1:200, Millipore, cat. no. MAB1281), MAP2 (1:200, Sigma, cat. no. M1406), Nestin (1:200, Abcam, cat. no. ab18102), NeuN (1:200, Sigma, cat. no. SAB5700017), NFM (1:200, Millipore, cat. no ab1987), PAX6 (1:50, Developmental Studies Hybridoma Bank, DSHB, cat. no. AB_528427), and SOX1 (1:200, R&D Systems, cat. no. AF3369). Bis-benzamide (DAPI, 1:1000, Sigma) was used to visualize the nuclei. Images were captured using a Zeiss AxioVision microscope (Zeiss, Oberkochen, German) with z-stack split view function or Keyence BZ-X800 (Keyence, Osaka, Japan).

### 4.11. Assessment of Spared White Matter by EC Staining

Spared myelinated white matter (WM) was quantified using a histological approach adapted from previously established protocols [13,67,72]. Tissue cassettes/sections were barcoded by a technician not involved in analysis. Imaging and quantification were performed blinded, using pre-specified settings. Briefly, every 10th spinal cord section (10 serial sections per set, spaced 200 µm apart) was stained with eriochrome cyanine to visualize preserved myelin. The lesion epicenter was defined as the section exhibiting the minimum amount of spared WM. Regions of white matter sparing were identified as areas with intact myelin structure and density, free of cystic cavities or degenerative changes. To assess the extent of WM sparing bilaterally, sections were analyzed at 400 µm intervals both rostrally and caudally from the epicenter, covering a total span of 2400 µm in each direction. The spared white matter volume was calculated using the formula volume = A × D, where A is the cross-sectional area of spared WM and D is the distance between measured sections (400 µm). Group differences in the spared areas at various distance from injury epicenter and the total spared WM volume were analyzed using one-way ANOVA, followed by Tukey’s post hoc test for pairwise comparisons.

## Figures and Tables

**Figure 1 ijms-26-08940-f001:**
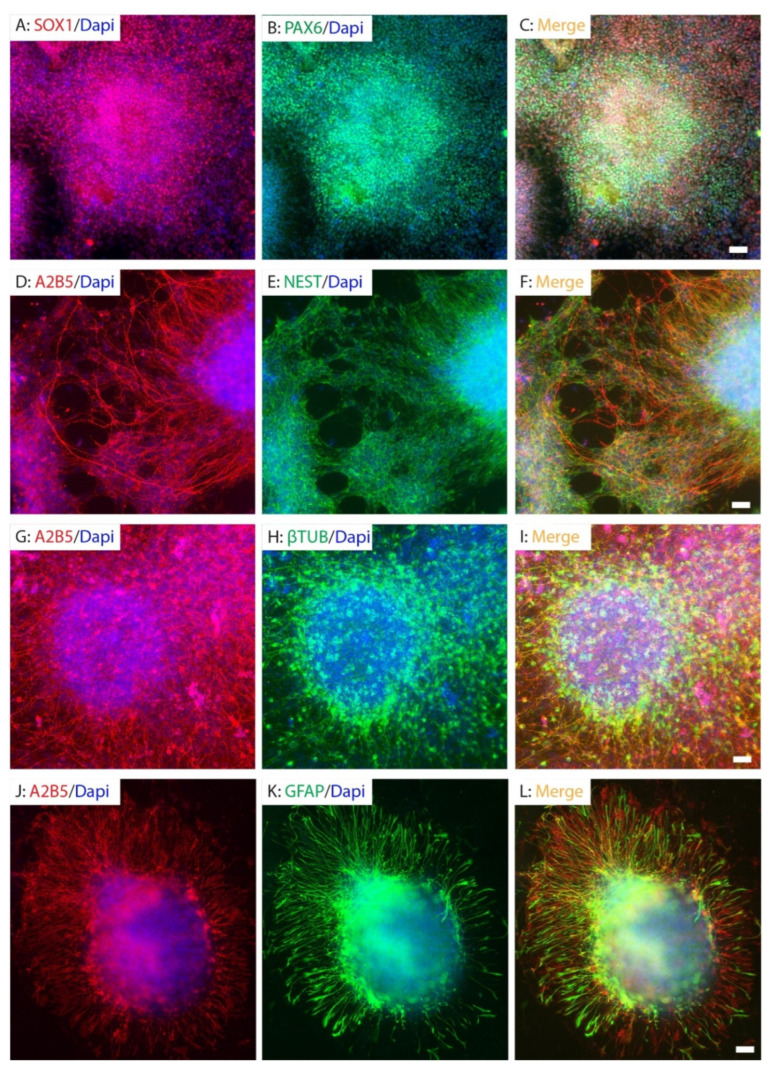
Neural differentiation of hiPSCs. The hiPSCs were differentiated into neural rosettes expressing neural stem cells markers SOX1 and PAX6 (**A**–**C**) at 15 days post-differentiation, NESTIN+ neural stem cells (**D**–**F**) and β-tubulin III+ neuronal progenitor cells (**G**–**I**) at 26 days PD, and GFAP+ glial precursor cells (**J**–**L**) at 40 days post-differentiation. Neural progenitor cells (NPCs) (**D**–**F**), neuronal progenitor cells (**G**–**I**), and glial progenitor cells (**J**–**L**) were all expressing A2B5. Scale bar = 50 µm.

**Figure 2 ijms-26-08940-f002:**
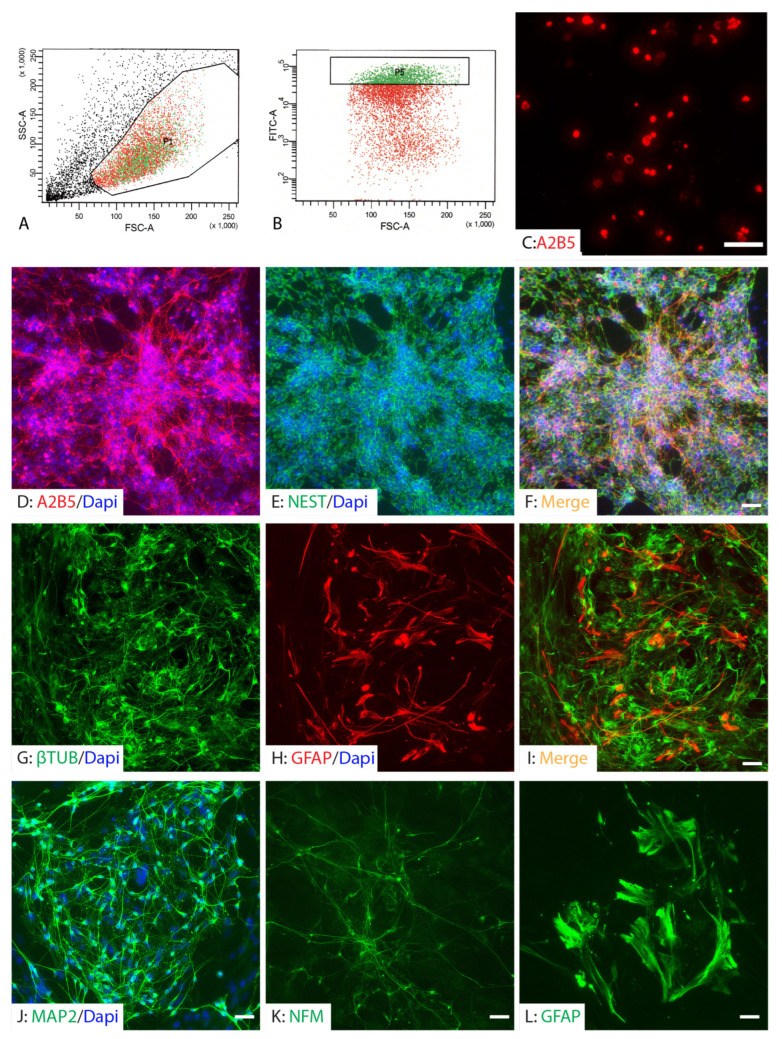
Purification and differentiation of iPSC-derived neural progenitor cells. iPSC-derived NPCs were sorted for A2B5+ cells (**A**,**B**), and only the top 15% with the highest A2B5 expression (cells box P5 in panel (**B**)) were collected for subsequent experiments to ensure purity. Almost all cells were A2B5+ at 1 day (**C**) after FACS. The purified A2B5+NPCs continued to proliferate in vitro in the presence of mitogens (**D**–**F**). The purified NPCs differentiated into neurons and astrocytes after withdrawal of mitogens for ten days (**G**–**I**). They continued to differentiate into mature neurons expressing MAP2 (**J**) and NFM (**K**) at 3 weeks in neuronal differentiation medium and mature astrocytes (**L**) at 4 weeks in the astrocyte differentiation medium. Scale bars = 50 µm.

**Figure 3 ijms-26-08940-f003:**
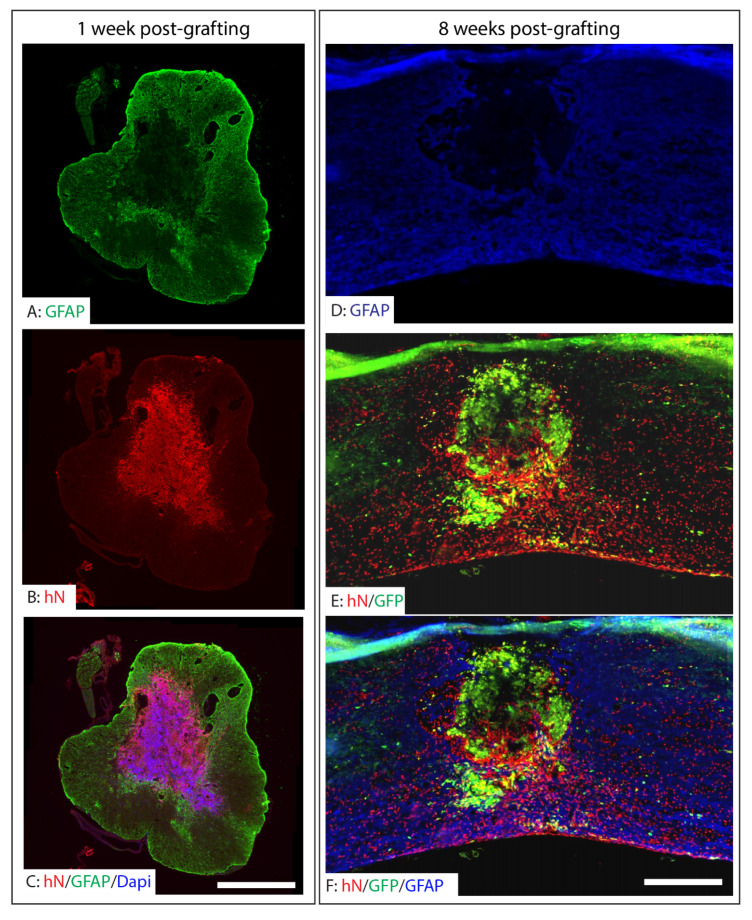
Survival and distribution of grafted hiPSC-derived neuronal progenitor cells in the injured spinal cord. The hiPSC-derived NPCs were transplanted at 8 days after T9 moderate contusion SCI in NOD-SCID mice. The grafted NPCs labeled by human nuclei survived in the injury epicenter at 1 week after transplantation (**A**–**C**). The grafted NPCs (**B**)were mainly located in the injured areas surrounded by the reactive astrocytes (**A**,**C**). The robust survival of grafted NPCs labeled by both human nuclei and GFP was observed in the injured spinal cord at 2 months after transplantation (**D**–**F**). While some grafted NPCs were located in the injury areas (**D**,**F**), many migrated into the spared spinal cord around the injury (**E**,**F**). Many grafted NPCs were labeled by human nuclei but not by GFP likely due to only partial NPCs being labeled by GFP before transplantation and some NPCs downregulating GFP expression after differentiation. Scale bars = 500 µm.

**Figure 4 ijms-26-08940-f004:**
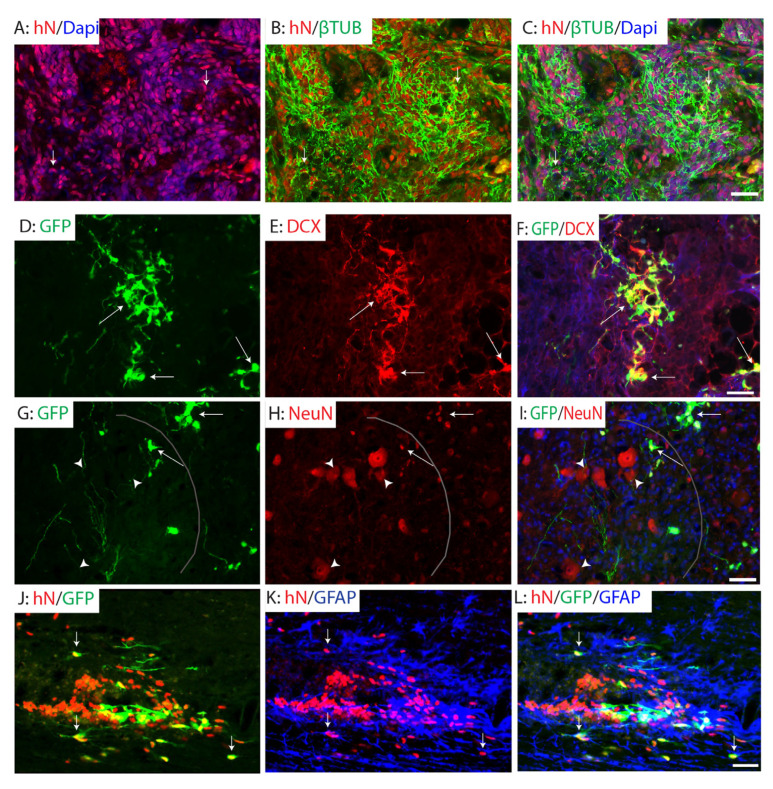
Differentiation of grafted NPCs after SCI. The grafted NPCs gradually matured into neurons over post-transplantation times. Many grafted NPCs were β-tubulin III+ neuronal progenitor cells at 1 week PD (arrows, (**A**–**C**)). They differentiated into immature doublecortin (DCX) at 2 weeks PD (arrows, (**D**–**F**)). At 2 months PD, majority of grafted NPCs differentiated into mature NeuN+ neurons in the spared gray matter (arrows, (**G**–**I**)). Importantly, GFP^+^ processes from graft-derived neurons extended into the spared ventral horn, where some were positioned in close proximity to host motor neurons and interneurons (arrowheads, (**G**,**I**)). Some grafted NPCs differentiated into GFAP+ astrocytes in the spared spinal cord around the injury (arrows, (**J**–**L**)). Scale bars = 50 µm.

**Figure 5 ijms-26-08940-f005:**
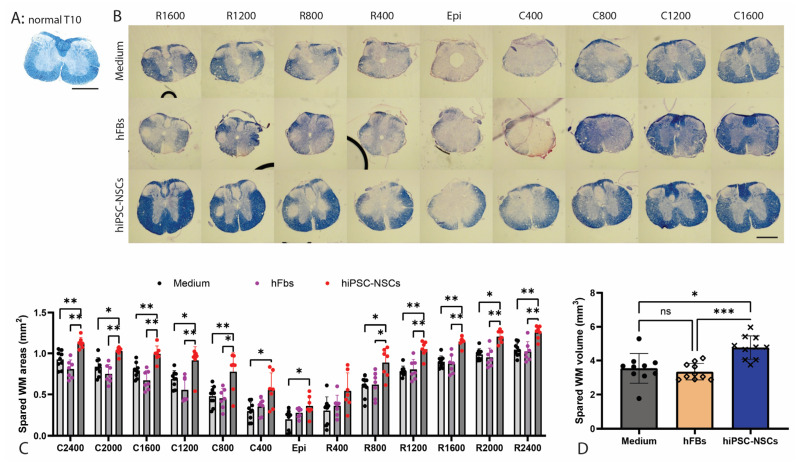
Spared white matter after transplantation of hiPSC-NPC. Normal uninjured white matter at T9 level with EC-staining is shown in (**A**). Spared white matter (WM) volumes in the injured side were quantified using EC staining at 2 months after transplantation (**B**). The spared WM areas were significantly greater in mice receiving NPC grafts compared to groups receiving human fibroblasts or medium (**C**). Similarly, the spared WM volumes were significantly greater in mice receiving NPC grafts compared to groups receiving human fibroblasts or medium (**D**). Scale bar = 500 μm (**A**). Data in (**C**,**D**) represent the mean ± SD (n = 10), * *p* < 0.05, ** *p* < 0.01, *** *p* < 0.001, ns: not significant (*p* > 0.05). Statistical analysis was performed using one-way ANOVA followed by Tukey’s post-test.

**Figure 6 ijms-26-08940-f006:**
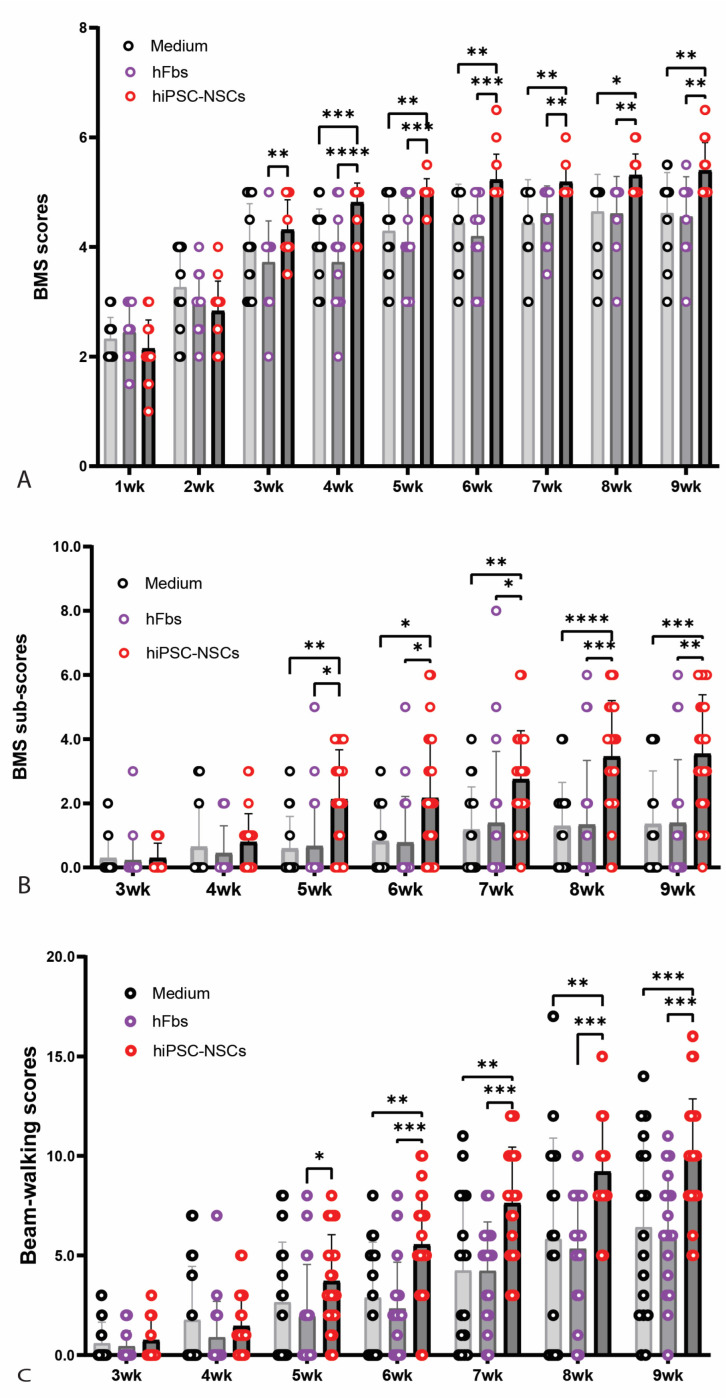
Locomotion recovery after transplantation of hiPSC-derived NPCs following SCI. Basso Mouse Scale (BMS) (**A**) were significantly greater in mice receiving NPC grafts compared to groups receiving human fibroblasts at 3 weeks after injury (2 weeks after transplantation) and in mice receiving NPC grafts compared to groups receiving human fibroblasts or culture medium starting 4 weeks after injury. BMS sub-scores (**B**) were also significantly greater in mice receiving NPC grafts compared to groups receiving human fibroblasts or culture medium starting 5 weeks after injury. Beam-walking scores (**C**) were significantly higher in NPC grafting group compared to hFb grafting group at 5 weeks after injury and in NPC group compared to other two groups starting 6 weeks injury. Data in A-C represent the mean ± SD (n = 17 in medium group, and 18 in hFB and NPC groups), * *p* < 0.05, ** *p* < 0.01, *** *p* < 0.001, **** *p* < 0.0001 Statistical analysis was performed using two-way ANOVA followed by Tukey’s post-test. The complete statistical analyses can be found in Appendix A.

**Table 1 ijms-26-08940-t001:** Sample sizes for functional and histology tests.

	Medium Ctrl	hFB Ctrl	hNPC Graft
Sample size by power analysis	N = 16	N = 16	N = 16
Exp. 1	N = 10	N = 10	N = 10
attrition	N = 2 ^a^	N = 1 ^b^	N = 1 ^c^
BMS	N = 8	N = 9	N = 9
Beam-walking	N = 8	N = 9	N = 9
IHC		N = 6	N = 9
EC staining	N = 5	N = 5	N = 5
Exp. 2	N = 10	N = 10	N = 10
attrition	N = 1 ^d^	N = 1 ^e^	N = 1 ^f^
BMS	N = 9	N = 9	N = 9
Beam-walking	N = 9	N = 9	N = 9
IHC		N = 3	N = 9
EC staining	N = 5	N = 5	N = 5
Final sample size for behavioral tests	N = 17	N = 18	N = 18
Final sample size for EC staining	N = 10	N = 10	N = 10

Note: ^a^ 1 mouse died during the first surgery and 1 mouse was sacrificed due to urinary infection after 2nd surgery; ^b^ mouse died during 2nd surgery; ^c^ infection and weight loss after 2nd surgery; ^d^ infection in the surgery site after 1st surgery; ^e^ urinary infection; ^f^ mouse died during 2nd surgery.

## Data Availability

The data presented in this study are available on request from the corresponding authors.

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
