# Peer review of "Functional Recovery by Transplantation of Human iPSC-Derived A2B5 Positive Neural Progenitor Cell After Spinal Cord Injury in Mice"

_ijms, 2025, doi:10.3390/ijms26188940_

Round 1
Reviewer 1 Report
Comments and Suggestions for Authors
The manuscript details the purification of A2B5⁺ neural progenitor cells (NPCs) from human induced pluripotent stem cells (hiPSCs), their characterization, and subsequent transplantation into a T9 contusion spinal cord injury model in NOD-SCID mice. The authors report survival, neuronal/astrocytic differentiation, increased spared white matter (as indicated by EC staining), and improved locomotor outcomes, without tumor formation, at 2 months post-grafting. The technical execution seems good, and the subject is both timely and clinically important.
The study shows promise, but it has some design and reporting issues that make it difficult to understand. Specifically, there is no A2B5⁻/unsorted NPC control, power/attrition reporting is missing, graft fate and integration are not fully quantified, and the safety window is too short. Taking care of these will make the paper much stronger.
Notes for the authors
1) A brief overview
You make hiPSC-derived NPCs, FACS-purify A2B5⁺ populations (with a purity threshold of 95% or higher), show in vitro bipotency, and transplant 1×10^5 cells into T9 contused NOD-SCID mice at 8 dpi. You report survival/migration, progressive neuronal maturation (βIII-tubulin → DCX → NeuN), and some astrocytic differentiation, as well as greater spared white matter and improved BMS/subscore and beam-walk outcomes compared to h-fibroblast or media controls at up to 2 months post-transplantation; no teratomas were observed.
2) Major comments
- Fullness of the control group.
The study does not include a control that distinguishes purification from transplantation itself. Please include (or explain the lack of) an unsorted NPC or A2B5⁻ (negative-sorted) cell group. This is important for testing the central claim that A2B5-based purification makes drugs more effective and safer than regular NPC grafting.
- Randomization, blinding, and loss of participants.
You specify group sizes (n = 10) and outline scoring methodologies. The manuscript, however, does not clearly state (i) the randomization method, (ii) how allocation was kept secret, (iii) how the experimenter was kept blind to histology/behavior, or (iv) how many people were left out or dropped out of the study (for example, deaths after an injury or mis-hits) for each group. To reduce bias and facilitate reproducibility, please include a CONSORT-style flow diagram and provide clear statements for each step, including the number of analyses performed for each endpoint.
- Why the sample size is what it is and what statistical assumptions are made.
Conduct a prospective power analysis for n=10, utilizing anticipated effect sizes for BMS/beam-walk/WM sparing. For a two-way repeated-measures ANOVA over 9 weeks, please provide the results of assumption checks (sphericity, normality, variance homogeneity) and any necessary corrections (e.g., Greenhouse-Geisser). For multiple comparisons, elucidate family-wise error control beyond Tukey at each time point. Please also include the exact p-values and effect sizes (η²/partial η², Cohen's d) along with 95% CIs.
- The magnitude and measurement of graft destiny.
Please provide stereological or density quantification of hN⁺ cells and their fates (NeuN⁺, DCX⁺, GFAP⁺) at defined distances from the epicenter, including numerator/denominator and areas sampled, as differentiation is shown qualitatively. Since only about 60% of cells were labeled with GFP, hN-based counts are necessary to avoid underestimating the number of cells due to loss or downregulation of the label. Please explain how you avoided counting bias and whether quantification was based on hN instead of GFP.
- The integration and function of neurons derived from grafts.
The text suggests axonal extension and connection with host neurons (Fig. 4), but more solid evidence is needed to support this. Think about adding:
- Synaptic markers (like human-specific Synapsin/PSD-95 next to ChAT/VGLUT/GAD terminals),
- Anterograde/retrograde tracing to show how the host and graft are connected,
- Or At the very least, figure out how many of the NeuN⁺ graft neurons are extending their axons into certain laminae.
The claims of "formed connection" seem too strong compared to the evidence provided, so they should be toned down or supported with numbers.
- Examination of white matter sparing.
EC-based WM quantification is suitable; however, please elucidate: (i) whether the analyses were conducted in a blinded manner, (ii) whether WM volumes were calculated bilaterally or solely on the “injured side,” and (iii) the objective definition of the lesion epicenter. In addition to total volume, include cross-sectional area data over distance (profiles) and give representative thresholding/segmentation criteria (or code) for reproducibility.
- Safety window and tumorigenicity.
Two months without teratomas is a positive sign, but it's not long enough because previous studies have shown that iPSC-derived grafts can take longer to grow. You should either extend the follow-up period to at least six to nine months or exercise greater caution in your conclusions. At the very least, you should include Ki-67 or another equivalent proliferation index and talk about the risks of long latency in the findings.
- Reporting on sex and age outside of the study.
Please provide the sex, age, and body weight of the mice, and indicate whether both sexes were included, as the results may vary by sex. Consider discussing the limitations of NOD-SCID hosts (immunodeficient milieu) and how this may exaggerate the appearance of graft survival and integration compared to immunocompetent models. Also, be honest about the gaps in translation.
- Information about the dose, timing, and viability.
You transplant 1×10^5 cells at 8 dpi; please explain why you made these choices (dose-response, timing in relation to the subacute microenvironment), report cell viability at the time of injection, and say whether you used multiple deposits or a single bolus. To reduce reflux and ensure more consistent procedures, include the coordinates, rate, and needle dwell time.
- Availability of data and openness of protocols.
Put in a statement about the availability of data and code, and (ideally) upload raw behavioral scores, EC segmentation masks, and analysis scripts. Thanks for the detailed list of antibodies and media. However, the catalog numbers are only partially provided. Please make sure that all reagents have sources/IDs for replication.
3) Minor comments (clarifications, presentation)
- Words and typos. In the figure legends, "man ± SD" should be used instead of "mean ± SD" throughout (Figs. 5–6).
- The size and visibility of the figure. Please provide us with panels of higher resolution, include scale bars in every subpanel, and clearly label regions and laminae (such as the ventral horn in Fig. 4).
- Reporting behavior in numbers. In addition to weekly plots, include a table of BMS/beam-walk means ± SD, n per time point, and exact p/CIs. Pre-injury baselines should be displayed to confirm equivalence at t=0.
- Factors in ANOVA. Clearly state the factors (Group × Time) and whether Time is within-subject; verify that the repeated-measures structure (animal as subject) was correctly modeled.
- Markers for imaging. Clearly state whether you used human-specific neuronal markers (such as hNCAM or STEM121) when double-labeling with NeuN/DCX to avoid confusion between the host and graft. Currently, you use hN/GFP in some instances; please standardize and quantify this with hN co-labeling.
- The effectiveness of GFP labeling. Since about 60% of the cells were GFP⁺, please make sure that all fate quantifications are based on hN and not GFP. If any counts were based on GFP, please make the necessary adjustments and report them or re-analyze them as needed.
- A statement of ethics. Animal welfare approval is noted; please include IACUC protocol numbers and pain relief information beyond the anesthetic regimen to meet ARRIVE standards.
- Updates to references. A few references are recent and appropriate; ensure consistency of formatting (journal abbreviations, DOIs) across the list and check for duplicates that appear in multiple sections.
4) Good points
- A purification strategy that is important for clinical use. FACS based on A2B5 is more likely to work in real life than reporter systems. The >95% post-sort purity standard is correct.
- A battery of tests on behavior and histology. Combining BMS, subscore, beam-walk, and EC myelin assessment yields converging endpoints; methodologies are delineated with practical operational specifics.
- A clear story of in-vivo maturation. The staged progression from βIII-tubulin to DCX to NeuN is demonstrated and aligns with expected timelines.
5) The range of claims
Please use more formal language when discussing circuit repair ("formed connection with motor and interneurons") unless you have synaptic or functional data (tracing, electrophysiology) to support it, as suggested above. Likewise, frame safety as “no macroscopic overgrowth/teratomas observed up to 2 months” instead of suggesting long-term safety.
6) Suggested additions (if possible during revision)
- Add an NPC comparator that is either negative-sorted (A2B5⁻) or not sorted.
- Offer quantitative fate mapping (hN-anchored) and limited synaptic marker analysis.
- For safety, extend observation to at least six months for a small group.
- Put raw data and analysis code in a repository and add a statement about Data Availability.
In conclusion, this is a promising and timely study with a reasonable experimental framework. The manuscript's impact and credibility will be significantly improved by addressing the control set, the rigor of the statistics/reporting, the quantitative integration/fate mapping, and the safety window.
Author Response
The manuscript details the purification of A2B5⁺ neural progenitor cells (NPCs) from human induced pluripotent stem cells (hiPSCs), their characterization, and subsequent transplantation into a T9 contusion spinal cord injury model in NOD-SCID mice. The authors report survival, neuronal/astrocytic differentiation, increased spared white matter (as indicated by EC staining), and improved locomotor outcomes, without tumor formation, at 2 months post-grafting. The technical execution seems good, and the subject is both timely and clinically important.
The study shows promise, but it has some design and reporting issues that make it difficult to understand. Specifically, there is no A2B5⁻/unsorted NPC control, power/attrition reporting is missing, graft fate and integration are not fully quantified, and the safety window is too short. Taking care of these will make the paper much stronger.
Notes for the authors
1) A brief overview
You make hiPSC-derived NPCs, FACS-purify A2B5⁺ populations (with a purity threshold of 95% or higher), show in vitro bipotency, and transplant 1×10^5 cells into T9 contused NOD-SCID mice at 8 dpi. You report survival/migration, progressive neuronal maturation (βIII-tubulin → DCX → NeuN), and some astrocytic differentiation, as well as greater spared white matter and improved BMS/subscore and beam-walk outcomes compared to h-fibroblast or media controls at up to 2 months post-transplantation; no teratomas were observed.
2) Major comments
Comments:
- Fullness of the control group.
The study does not include a control that distinguishes purification from transplantation itself. Please include (or explain the lack of) an unsorted NPC or A2B5⁻ (negative-sorted) cell group. This is important for testing the central claim that A2B5-based purification makes drugs more effective and safer than regular NPC grafting.
Responses:
Thank the reviewer for the suggestion. We did not include an unsorted NPC group, as prior studies have shown that unsorted NPC grafts frequently lead to tumor-like overgrowth beginning around 6 weeks post-transplantation (refs 32, 40, 49, 50). Similarly, we excluded the A2B5-negative group for two reasons: (1) it contains mixed populations, including undifferentiated iPSCs that inevitably cause tumor formation after grafting; and (2) the heterogeneity of these cells makes it difficult to directly compare therapeutic efficacy with the lineage-restricted A2B5⁺ NPCs. Therefore, we focused on characterizing A2B5-positive sorted cells for grafting and subsequent experiments. We have added the explanation to the Materials and Methods section, under Section 4.7. Thoracic contusion SCI and cell transplantation.
Comments:
- Randomization, blinding, and loss of participants.
You specify group sizes (n = 10) and outline scoring methodologies. The manuscript, however, does not clearly state (i) the randomization method, (ii) how allocation was kept secret, (iii) how the experimenter was kept blind to histology/behavior, or (iv) how many people were left out or dropped out of the study (for example, deaths after an injury or mis-hits) for each group. To reduce bias and facilitate reproducibility, please include a CONSORT-style flow diagram and provide clear statements for each step, including the number of analyses performed for each endpoint.
Responses:
Thank the reviewer for the suggestion. We have added a subsection 4.7. Animals to describe in detail on randomization, allocation concealment, blinding, and attrition. We have also included Table 1 showing the number of animals used for each test and attrition.
Comments:
- Why the sample size is what it is and what statistical assumptions are made.
Conduct a prospective power analysis for n=10, utilizing anticipated effect sizes for BMS/beam-walk/WM sparing. For a two-way repeated-measures ANOVA over 9 weeks, please provide the results of assumption checks (sphericity, normality, variance homogeneity) and any necessary corrections (e.g., Greenhouse-Geisser). For multiple comparisons, elucidate family-wise error control beyond Tukey at each time point. Please also include the exact p-values and effect sizes (η²/partial η², Cohen's d) along with 95% CIs.
Responses:
The sample size was determined by a priori power analysis using G*power software (N=16 per group, 3 groups total, based on alpha value of 0.01 (chosen conservatively), 0.5 effective size, 0.7 correlation among repeated measures resulting in a power of 0.83). The detailed statistical analysis using a two-way repeated-measures ANOVA over 9 weeks has been included as the Supplementary Files 1, 2 and 3.
Comments:
- The magnitude and measurement of graft destiny.
Please provide stereological or density quantification of hN⁺ cells and their fates (NeuN⁺, DCX⁺, GFAP⁺) at defined distances from the epicenter, including numerator/denominator and areas sampled, as differentiation is shown qualitatively. Since only about 60% of cells were labeled with GFP, hN-based counts are necessary to avoid underestimating the number of cells due to loss or downregulation of the label. Please explain how you avoided counting bias and whether quantification was based on hN instead of GFP.
Responses:
We agree that these quantifications are important preclinical data for advancing NPC transplantation toward clinical translation. This study represents an initial step in our effort to bring hiPSC-derived NPCs to the bedside, focusing on the therapeutic potential of A2B5⁺ NPC grafts for locomotor recovery and their early tumorigenic risk. In follow-up studies already planned, we will evaluate long-term safety (up to one and half year post-transplantation) and investigate the mechanisms by which NPC grafts promote functional recovery, including the detailed quantifications suggested. These extended studies will provide additional time points to more precisely characterize the temporal dynamics of NPC differentiation in vivo.
Comments:
- The integration and function of neurons derived from grafts.
The text suggests axonal extension and connection with host neurons (Fig. 4), but more solid evidence is needed to support this. Think about adding:
- Synaptic markers (like human-specific Synapsin/PSD-95 next to ChAT/VGLUT/GAD terminals),
- Anterograde/retrograde tracing to show how the host and graft are connected,
- Or At the very least, figure out how many of the NeuN⁺ graft neurons are extending their axons into certain laminae.
The claims of "formed connection" seem too strong compared to the evidence provided, so they should be toned down or supported with numbers.
Responses:
Thank the reviewer for the comments. 1). We have tried several antibodies for synapsin and PSD-95. Unfortunately, none is human specific. 2). The trans-synaptic viral tracing experiments will be done in the following up studies. 3). The grafted neurons were not restricted to the injury epicenter but also migrated into spared regions of the spinal cord. Therefore, the presence of graft-derived GFP⁺ processes within specific laminae could reflect projections from grafted neurons located in other laminae and/or the migration of grafted neurons directly into those laminae.
Thus, we agreed with this reviewer to tone down the statement of “formed connection” between host and grafted neurons. In “Result 2.4” as well as “Discussion”, we have revised as the following: Notably, GFP⁺ processes from graft-derived neurons extended into the spared ventral horn, where some were positioned in close proximity to host motor neurons and interneurons. These findings suggest the potential for grafted neurons to form connections with host neurons. However, further studies are required to validate synaptic integration of grafted neurons into host circuits, for example using immuno-electron microscopy or transsynaptic viral tracing.
Comments:
- Examination of white matter sparing.
EC-based WM quantification is suitable; however, please elucidate: (i) whether the analyses were conducted in a blinded manner, (ii) whether WM volumes were calculated bilaterally or solely on the “injured side,” and (iii) the objective definition of the lesion epicenter. In addition to total volume, include cross-sectional area data over distance (profiles) and give representative thresholding/segmentation criteria (or code) for reproducibility.
Responses:
1) The analyses of spared white matter were conducted by a researcher who is blind to the experiment groups. 2) We used the T10 central contusion which caused injury in both sides. WM volumes were calculated bilaterally. 3) The lesion epicenter was defined as the section exhibiting the minimum amount of spared WM. 4) Cross-sectional area data over distance is shown in Figure 5, where representative images from epicenter, rostral direction from 400 µm to 1600 µm, and caudal direction from 400 µm to 1600 µm are included. The spared WM areas over distance from the injury epicenter are now included in revised Figure 5C. Description of EC-based WM quantification is included in Section 4.11.
Comments:
- Safety window and tumorigenicity.
Two months without teratomas is a positive sign, but it's not long enough because previous studies have shown that iPSC-derived grafts can take longer to grow. You should either extend the follow-up period to at least six to nine months or exercise greater caution in your conclusions. At the very least, you should include Ki-67 or another equivalent proliferation index and talk about the risks of long latency in the findings.
Responses:
We agree with the reviewer that to examine tumor formation, longer safety window is needed, which is part of our ongoing project. We have emphasized that (in Discussion section) “...further studies are needed to identify the markers for the tumorigenic NSCs in the injured spinal cord after transplantation and to clarify their relationship with the highly proliferative A2B5 negative NSCs. In addition, long-term studies are essential to further validate the safety of A2B5+ NPC transplantation.”
Comments:
- Reporting on sex and age outside of the study.
Please provide the sex, age, and body weight of the mice, and indicate whether both sexes were included, as the results may vary by sex. Consider discussing the limitations of NOD-SCID hosts (immunodeficient milieu) and how this may exaggerate the appearance of graft survival and integration compared to immunocompetent models. Also, be honest about the gaps in translation.
Adult female NOD-SCID mice (3–6 months old, 20-40 g) were used in our experiments. We have specified this information in Section 4.7.
Comments:
- Information about the dose, timing, and viability.
You transplant 1×10^5 cells at 8 dpi; please explain why you made these choices (dose-response, timing in relation to the subacute microenvironment), report cell viability at the time of injection, and say whether you used multiple deposits or a single bolus. To reduce reflux and ensure more consistent procedures, include the coordinates, rate, and needle dwell time.
Responses:
Information about the dose, timing, and viability is described in Section 4.7. The volume of injection, total cell number and timing to the subacute microenvironment were chosen based on our previous experience and publications (Cao, et al, 2010, J Neurosci, doi:10.1523/jneurosci.3174-09.2010). Briefly, 2 μL cell suspension (total of 10^5 cells) were injected into the injury epicenter at a depth of 0.7 mm using a glass micropipette (outer diameter 50–70 μm, tip beveled to 30–50°), at a controlled rate of 0.2 μL/min, for a total dwell time of 5 min as previously described.
Comments:
- Availability of data and openness of protocols.
Put in a statement about the availability of data and code, and (ideally) upload raw behavioral scores, EC segmentation masks, and analysis scripts. Thanks for the detailed list of antibodies and media. However, the catalog numbers are only partially provided. Please make sure that all reagents have sources/IDs for replication.
Responses:
Thank the reviewer for the suggestions. We have added Data Availability Statement at the end of the manuscript. Data Availability Statement: The data presented in this study are available on request from the corresponding authors. Catalog numbers for all antibodies are now included in the Material and Methods section (Section 4.10 Immunofluorescence and immunohistochemistry).
Comments:
3) Minor comments (clarifications, presentation)
- Words and typos. In the figure legends, "man ± SD" should be used instead of "mean ± SD" throughout (Figs. 5–6).
Responses:
Thank the reviewer for pointing out. The typos are now corrected.
Comments:
The size and visibility of the figure. Please provide us with panels of higher resolution, include scale bars in every subpanel, and clearly label regions and laminae (such as the ventral horn in Fig. 4).
Responses:
The images in figure 4G-I were acquired at high magnification, with a reference line added to indicate boundary between laminae IX and VIII in ventral horn.
Comments:
- Reporting behavior in numbers. In addition to weekly plots, include a table of BMS/beam-walk means ± SD, n per time point, and exact p/CIs. Pre-injury baselines should be displayed to confirm equivalence at t=0.
Responses:
The detailed statistical analyses are now added as the Supplementary Files 1, 2 and 3. All animals used for functional assessments had baseline BMS and beam-walk tests before injury. Only animals with the normal BMS score (9) and beam walk score (25) were used for this study.
Comments:
- Factors in ANOVA. Clearly state the factors (Group × Time) and whether Time is within-subject; verify that the repeated-measures structure (animal as subject) was correctly modeled.
Responses:
The detailed statistical analyses are now added as the Supplementary Files 1, 2 and 3.
Comments:
- Markers for imaging. Clearly state whether you used human-specific neuronal markers (such as hNCAM or STEM121) when double-labeling with NeuN/DCX to avoid confusion between the host and graft. Currently, you use hN/GFP in some instances; please standardize and quantify this with hN co-labeling.
Responses:
We used human nuclei (hN) to label human cells and used GFP to trace transplanted human cells. We used hN for quantification. We did not use hNCAM or STEM121 in our study.
Comments:
- The effectiveness of GFP labeling. Since about 60% of the cells were GFP⁺, please make sure that all fate quantifications are based on hN and not GFP. If any counts were based on GFP, please make the necessary adjustments and report them or re-analyze them as needed.
Responses:
We agree with the reviewer that only hN should be used for quantification of grafted NPCs.
Comments:
- A statement of ethics. Animal welfare approval is noted; please include IACUC protocol numbers and pain relief information beyond the anesthetic regimen to meet ARRIVE standards.
Responses:
We have added Approved animal protocol information. Pain relief information was added to Section 4.7. After injury animals were monitored, fed with presoftened food more palatable and receive prophylactic against pain. Postoperative care included subcutaneous administration of buprenorphine (0.05 mg/kg, twice daily for three days; Reckitt Benckiser, Hull, England) and manually emptying the bladder twice a day until the automatic voiding returned spontaneously.
Comments:
- Updates to references. A few references are recent and appropriate; ensure consistency of formatting (journal abbreviations, DOIs) across the list and check for duplicates that appear in multiple sections.
Responses:
We have checked and corrected references.
Comments:
4) Good points
- A purification strategy that is important for clinical use. FACS based on A2B5 is more likely to work in real life than reporter systems. The >95% post-sort purity standard is correct.
- A battery of tests on behavior and histology. Combining BMS, subscore, beam-walk, and EC myelin assessment yields converging endpoints; methodologies are delineated with practical operational specifics.
- A clear story of in-vivo maturation. The staged progression from βIII-tubulin to DCX to NeuN is demonstrated and aligns with expected timelines.
5) The range of claims
Please use more formal language when discussing circuit repair ("formed connection with motor and interneurons") unless you have synaptic or functional data (tracing, electrophysiology) to support it, as suggested above. Likewise, frame safety as “no macroscopic overgrowth/teratomas observed up to 2 months” instead of suggesting long-term safety.
Responses:
As mentioned above, we have toned down the claim of connection between the grafted and host neurons. We also revised the claim of frame safety as suggested in our discussion.
Comments:
6) Suggested additions (if possible during revision)
- Add an NPC comparator that is either negative-sorted (A2B5⁻) or not sorted.
- Offer quantitative fate mapping (hN-anchored) and limited synaptic marker analysis.
- For safety, extend observation to at least six months for a small group.
- Put raw data and analysis code in a repository and add a statement about Data Availability.
In conclusion, this is a promising and timely study with a reasonable experimental framework. The manuscript's impact and credibility will be significantly improved by addressing the control set, the rigor of the statistics/reporting, the quantitative integration/fate mapping, and the safety window.
Responses:
Thank the reviewer for the suggestions. We have revised the manuscript accordingly and added Supplementary Files 1, 2 and 3 for statistical analysis.

Reviewer 2 Report
Comments and Suggestions for Authors
This manuscript concerns the use of transplanted neural progenitor cells to treat spinal cord injury. It is straightforward and well-presented. There are, however, some minor details that could improve the manuscript.
- Detail should be added about the mouse husbandry conditions, including the source of the colony (in-house or purchased- a reference for this strain would also be helpful),
- More detail on the lentivirus-GFP used to label transplanted cells would be helpful. What promoter was used? Does the virus label only neurons, or other cells as well?
- Similarly, pre-implantation labeling with lentivirus would helpful to confirm the number of cells transduced with the virus.
- The FACS sorting data should be presented, not just micrographs before and after.
- Why bother to look at astrocyte differentiation under glial specific conditions if those cells aren't specifically transplanted. When the authors look at astrocytes after transplantation, they ostensibly differentiated from the same NPCs as the neurons.
- For Figure 5, uninjured spinal cords should be shown as controls for the normal level of white matter.
- For Figure 6, individual points should be graphed instead of bar graphs, similar to Figure 5B.
- Did the authors ever verify neuronal / astrocyte function in the differentiated iPSCs? The included data only confirms morphology and the expression of certain markers.
- Figure 2B-C- where is the beta-tubulin staining for these cells post-FACS? This would be a more direct comparison to Fig. 2A.
- Please make clearer that the GFAP-reactive astrocytes used to visualize the spinal cord are not from the transplantation (FIgure 3A / D)
- It difficult to visually confirm that the arrowheads in Figure 4G-I truly signify that the grafted neuron are making connections with motor and interneurons.
Author Response
This manuscript concerns the use of transplanted neural progenitor cells to treat spinal cord injury. It is straightforward and well-presented. There are, however, some minor details that could improve the manuscript.
Comments:
- Detail should be added about the mouse husbandry conditions, including the source of the colony (in-house or purchased- a reference for this strain would also be helpful),
Responses:
Thank the reviewer for the suggestion. We have added husbandry information and the source of the colony in Section 4.7: Adult NOD-SCID mice were purchased from the Jackson Laboratory, NOD.Cg-Prkdc scid/J, strain no. 001303.
Comments:
- More detail on the lentivirus-GFP used to label transplanted cells would be helpful. What promoter was used? Does the virus label only neurons, or other cells as well?
Responses:
Thank the reviewer for the suggestion. The lentivirus GFP is used to label NPCs. The GFP expression is driven by a constitutive CMV promoter and will label NPCs and their derivatives, including neurons. We have added a new Figure Supplementary Figure S1 to show details of the lentiviral vector information.
Comments:
- Similarly, pre-implantation labeling with lentivirus would helpful to confirm the number of cells transduced with the virus.
Responses:
About 60% of our NPCs can be labeled with lentiviral GFP before transplantation.
Comments:
- The FACS sorting data should be presented, not just micrographs before and after.
Responses:
Thank the review for the suggestion. The FACS data are now included in Figure 2 A, B.
Comments:
- Why bother to look at astrocyte differentiation under glial specific conditions if those cells aren't specifically transplanted. When the authors look at astrocytes after transplantation, they ostensibly differentiated from the same NPCs as the neurons.
Responses:
We examined astrocyte differentiation to confirm that A2B5+ sorted NPCs have the capacity to generate astrocytic as well as neuronal lineages. The in vivo findings are consistent with our in vitro data (Figure 1), showing that A2B5 is expressed in both neuronal- and glial-precursor cells. These precursors likely differentiate into neurons and astrocytes, both of which may be required for functional recovery, as discussed in detail in the section of discussions.
Comments:
- For Figure 5, uninjured spinal cords should be shown as controls for the normal level of white matter.
Responses:
Thank the reviewer for the suggestion. The uninjured spinal cord at T9 is now included. Figure 5 shows spared white matter from a full range cross-sections of rostral to caudal cord after injury at the epicenter (T9 level), as well as the spared white matter volume from three groups: A2B5+ NPC, human fibroblasts, and culture medium.
Comments:
- For Figure 6, individual points should be graphed instead of bar graphs, similar to Figure 5B.
Responses:
Thank the reviewer for the suggestion. Figure 6 is revised accordingly.
Comments:
- Did the authors ever verify neuronal / astrocyte function in the differentiated iPSCs? The included data only confirms morphology and the expression of certain markers.
Responses:
The functions of neurons and astrocytes derived from differentiated iPSCs were assessed collectively through behavioral assays and functional recovery tests following grafting. We did not conduct separate evaluations to distinguish the specific contributions of individual neuronal or astrocytic subtypes.
Comments:
- Figure 2B-C- where is the beta-tubulin staining for these cells post-FACS? This would be a more direct comparison to Fig. 2A.
Responses:
Figure 2 is revised according per this reviewer’s suggestion. The original 2A and 2C are partially overlapped with Figure 1G and 2F, respectively, and are now deleted. The FACS data are now shown in the revised Figure 2A and B. The revised Figure 2C is the staining for living cell showing the survival and purity of sorted A2B5+NPCs. The beta-tubulin staining of cells post-FACS is shown in Figure 2G.
Comments:
- Please make clearer that the GFAP-reactive astrocytes used to visualize the spinal cord are not from the transplantation (FIgure 3A / D)
Responses:
The GFAP immunostaining is used to show the injury site (GFAP negative areas). The GFAP-reactive astrocytes used to visualize the spinal cord are not from the transplantation because Figure 3A is one week post transplantation and at that stage, grafted cells are still at the NPC stage and would not be expressing GFAP. Even at 8 weeks post transplantation (Figure 3D), the number of human cells grafted would not be repopulated to the entire spinal cord.
Comments:
- It difficult to visually confirm that the arrowheads in Figure 4G-I truly signify that the grafted neuron are making connections with motor and interneurons.
Responses:
Thank the reviewer for pointing this out. We have moved the arrowheads close to host neurons which appear to receive the GFP+ projection derived from grafted neurons. We also toned down the claim of “connection between grafted and host neurons”.

Round 2
Reviewer 1 Report
Comments and Suggestions for Authors
The amended manuscript adressed many of the reporting/statistical issues, but some of the requested reviewer's core experimental request are not fully satisfied such as controls, quantitative fate mapping, extended safety. The response to reviewer comments are partially fullfilled.
Author Response
Comments: The amended manuscript adressed many of the reporting/statistical issues, but some of the requested reviewer's core experimental request are not fully satisfied such as controls, quantitative fate mapping, extended safety. The response to reviewer comments are partially fullfilled.
Responses: We sincerely thank the reviewer for the timely response and thoughtful feedback. We fully agree that the points raised—including appropriate controls, quantitative fate mapping, and extended safety evaluation—are critical for the translational advancement of hiPSC-NPC therapies, as emphasized in our earlier responses. At the same time, we believe it is essential to first establish the therapeutic potential of hiPSC-derived A2B5⁺ NPCs for functional recovery, which is the primary focus of the present study, before undertaking these important but time- and labor-intensive follow-up studies.
Our findings provide strong evidence that A2B5⁺ NPCs hold great promise for SCI treatment. These results are particularly significant given that a clinical trial using hiPSC-NPCs for SCI has already been approved. Incorporating A2B5-based FACS into cGMP-compliant protocols is feasible and could generate safer, clinical-grade NPCs for therapeutic use. Importantly, the positive functional outcomes reported here further justify the need for the additional studies suggested by the reviewer, which we have already planned for future work.
Reviewer 2 Report
Comments and Suggestions for Authors
The authors have adequately addressed the concerns of the reviewer.
Author Response
Comments: The authors have adequately addressed the concerns of the reviewer.
Response: We appreciate the reviewer’s positive comments and confirmation that the concerns have been addressed.